# Low-Density Lipoprotein Pathway Is a Ubiquitous Metabolic Vulnerability in High Grade Glioma Amenable for Nanotherapeutic Delivery

**DOI:** 10.3390/pharmaceutics15020599

**Published:** 2023-02-10

**Authors:** Adenike O. Adekeye, David Needham, Ruman Rahman

**Affiliations:** 1Biodiscovery Institute, School of Medicine, University of Nottingham, Nottingham NG7 2RD, UK; 2Department of Mechanical Engineering and Material Science, School of Engineering, Duke University, Durham, NC 27708, USA

**Keywords:** low-density lipoprotein, high grade glioma, perivascular, nanotherapy

## Abstract

Metabolic reprogramming, through increased uptake of cholesterol in the form of low-density lipoproteins (LDL), is one way by which cancer cells, including high grade gliomas (HGG), maintain their rapid growth. In this study, we determined LDL receptor (LDLR) expression in HGGs using immunohistochemistry on tissue microarrays from intra- and inter tumour regions of 36 adult and 133 paediatric patients to confirm LDLR as a therapeutic target. Additionally, we analysed expression levels in three representative cell line models to confirm their future utility to test LDLR-targeted nanoparticle uptake, retention, and cytotoxicity. Our data show widespread LDLR expression in adult and paediatric cohorts, but with significant intra-tumour variation observed between the core and either rim or invasive regions of adult HGG. Expression was independent of paediatric tumour grade or identified clinicopathological factors. LDLR-expressing tumour cells localized preferentially within perivascular niches, also with significant adult intra-tumour variation. We demonstrated variable levels of LDLR expression in all cell lines, confirming their suitability as models to test LDLR-targeted nanotherapy delivery. Overall, our study reveals the LDLR pathway as a ubiquitous metabolic vulnerability in high grade gliomas across all ages, amenable to future consideration of LDL-mediated nanoparticle/drug delivery to potentially circumvent tumour heterogeneity.

## 1. Introduction

Gliomas are the most common primary brain tumours and originate from the glial cells of the brain. They are a heterogenous spectrum of slow-growing to highly aggressive infiltrating tumours and have been classified based on their histological features into four grades by the World Health Organization (WHO); grade 1 (pilocytic astrocytoma), grade 2 (diffuse astrocytoma), grade 3 (anaplastic astrocytoma), and grade 4 (glioblastoma, GBM) [1]. Grades 3 and 4 constitute nearly half of all malignant primary brain tumours and are collectively referred to as high grade gliomas (HGG) due to their highly aggressive nature and dismal prognosis. GBM in particular is one of the most aggressive and treatment-resistant cancers in adults and children alike with a median survival of 4.6 months in the absence of treatment and approximately 14 months with optimal multimodal treatment, with a 5-year survival rate of less than 5% and over 20 average years of life lost signifying the devastating burden it poses [2,3,4,5,6,7].

Since the mid-twentieth century, standard-of-care treatment has involved surgical resection of the tumour followed by external beam radiotherapy (except in children under 3 years) and more recently the use of adjuvant chemotherapeutic agents such as temozolomide, nitrosoureas and polymer-based carmustine (Gliadel^®^ wafer) placed at the affected tumour site during surgery [3,8,9,10]. However, despite these advances in treatment methods, the near-impossibility of complete tumour removal, poor drug delivery to the tumour regions due to the blood–brain barrier (BBB), and the debilitating adverse effects of treatment especially in children, have all contributed to the failure in significant outcome improvement, as even with the best of treatment, HGGs recur in over 90% of cases [3,10].

Over the past decade, there has been increasing evidence to suggest that intra-tumour heterogeneity (different sub-clonal cell populations with different molecular profiles within the same tumour) exists in the majority of cancers including malignant brain tumours, and it is in fact fundamental to our understanding of the variation in patient response to treatment and treatment failure [3]. Studies have also shown that although histologically similar, paediatric, and adult brain tumours in-fact differ in their genetic alterations and pathologic features with paediatric tumours presenting fewer genetic mutations. This discovery throws in doubt the effectiveness of current standard-of-care drugs administered to children which were originally developed for use in their adult counterparts. Hence, there is increasing interest in finding a common, ubiquitous vulnerability that is prevalent across the various intra-tumour sub-clones, disregarding the heterogeneity and molecular differences between paediatric and adult tumour types towards which novel therapeutic strategies can be targeted [11,12,13,14].

Abnormal cellular metabolism is one of the hallmarks of cancer, with most cancer cells including HGGs relying on glycolysis rather than oxidative phosphorylation for energy supply, a phenomenon known as the Warburg effect. Additionally, the increased requirement for cholesterol to maintain tumour proliferation and spread is an observed feature in many cancers [15,16,17]. Cholesterol is made available to tumour cells either through; (i) increased cellular synthesis regulated by the enzymes fatty acid synthase (FASN), which catalyses the conversion of acetyl-CoA and malonyl-CoA to palmitate and 3-hydroxyl-3-methylglutaryl-coenzyme A (HMG-CoA) reductase or (ii) from cellular uptake of preformed small-sized (20–25 nm) low-density lipoproteins (LDL), which bind to the high affinity LDL receptors (LDLR) found in clathrin-coated pits on the cell surface, which are internalised into the cytoplasm via receptor-mediated endocytosis as lipid vesicles and then transported to lysosomes, where they are hydrolysed for subsequent use by the cells and the dissociated receptors recycled to the cell surface for more LDL uptake [15,16,18,19,20,21].

Overexpression of FASN is associated with various pathological conditions such as cardiovascular diseases, diabetes and several different cancers including gliomas, breast, prostate, lung and colon cancer [15]. Similarly, LDLR has been shown to be upregulated by a factor of hundreds in rapidly proliferating tumours such as leukaemias, gynaecological malignancies, lung cancer, and across numerous human glioma cell lines compared to their respective normal tissues [16,17,20]. Due to this pro-tumorigenic effect, LDLR expression has been shown to be a predictor of tumour progression and survival with studies of breast cancer and small-cell lung cancer patients exhibiting better survival rates in those whose tumours demonstrated low LDLR expression [22,23]. Although the extent to which tumour cells including malignant gliomas meet their cholesterol requirement is not fully understood, FASN and LDLR are potential targets for therapeutic intervention to circumvent tumour heterogeneity in HGGs [16,20]. One such intervention, described by Lacko et al., is the creation of synthetic nano-LDL particles within which the chemotherapeutic agent is encapsulated to serve as drug delivery vehicles, targeting the LDLRs, or where the drug is designed to be of similar size to LDL so that they are preferentially taken up by the cancer cells as they would for normal LDL- effectively the cancer cell’s “food” [11,20,24,25]. A new improvement to this method is the synthesis of dual-targeting nanoparticles directed at both the tumour and BBB such that the nanoparticle can access the tumour irrespective of whether or not the BBB is intact [26]. The ability of LDL nanoparticle-delivery systems to escape immune detection, due to its properties as a natural molecule, has made it a better prospect compared to other delivery systems and also of potential use in other cancer types such as breast cancer [27].

Here, we test the hypothesis that invariably high levels of expression of LDLR protein across intra- and inter-tumour regions in adult and paediatric high-grade gliomas confirm LDLR as a therapeutic target. This was determined through immunohistochemistry (IHC) using an LDL-receptor targeting antibody carried out on patient-derived tissue microarrays (TMA), which incorporated varying intra-tumour regions in the adult GBM cohort and single region samples for the paediatric HGGs. Additionally, representative cell lines of adult (GIN28) and paediatric (SF188, KNS42) glioblastoma were assessed for LDLR expression by immunofluorescence (IF) in vitro to confirm the future utility of these cell lines as models to test fluorescently labelled LDLR-targeted stealth nanoparticle uptake, retention, and cytotoxicity.

## 2. Materials and Methods

### 2.1. Tissue Samples and Cell Lines

Tissue microarrays (TMAs) were previously prepared in the Children’s Brain Tumour Research Centre (CBTRC) from adult and paediatric patients with confirmed diagnosis of high-grade glioma (HGG) from multi-centre trials, retrospective studies and cases registered with the Children’s Cancer and Leukaemia Group (CCLG). A total of 21 paediatric TMAs, each consisting of cores from different areas of a single biopsy specimen and 4 adult TMAs containing cores taken from three distinct intra-tumour regions were studied, with each TMA housing samples from 10–27 patients in triplicates. There was a total of 36 adult GBM and 133 paediatric high grade glioma cases, of which clinical diagnosis was available for 68 of paediatric cases. Of these, 13 were grade III and 55 grade IV.

Primary glioblastoma multiforme (GBM) cell line (GIN28) was generated from invasive tumour margin of adult brain tumour patients treated at the Nottingham University Hospitals (NUH) while, paediatric GBM cell lines (SF188, KNS42) were obtained from Professor Chris Jones of the Institute of Cancer Research (ICR), London, UK.

This project was approved by the National Research Ethics Committee (NRES Committee East Midlands) and the granted Ethics Reference Number is 11/EM/0076.

### 2.2. Immunohistochemistry (IHC)

Low density lipoprotein receptor (LDLR) protein in human HGG samples was detected using indirect IHC for better signal amplification. Formalin-fixed paraffin-embedded (FFPE) slide sections were deparaffinized in xylene for 15 min and rehydrated in decreasing concentrations of alcohol and running water. Slides were placed in a pre-heated Sodium Citrate buffer solution (pH 6.0) for 40 min in a steamer for antigen retrieval, cooled for 10–20 min, and washed in phosphate-buffered saline (PBS). All incubations were carried out in a humidified box at room temperature and washes between stages in PBS. Sections were covered in 20% normal goat serum (NGS) in PBS for 20 min to minimize non-specific antibody binding and endogenous peroxidase activity was blocked by incubating slides in DAKO peroxidase blocking solution for 5 min to prevent background staining. The primary antibody used was Abcam (Biomedical Campus, Discovery Drive, Trumpington, Cambridge, CB2 0AX, UK) anti-LDLR rabbit monoclonal antibody (EP1553Y) ab52818 and was optimized on normal liver tissue (positive control) at dilutions ranging from 1:250 to 1:1400 incubated for 1 h at room temperature, subsequent experiments were performed at 1:1400 dilution. Slides were incubated in DAKO REAL Envision HRP Rabbit/Mouse secondary antibody for 30 min. Detection of antibody binding was enzyme-based following incubation in DAKO diaminobenzindine (DAB) chromogen substrate for 10–15 min. Slides were counterstained in Gills Haematoxylin, differentiated in lithium carbonate solution, washed in running water, and dehydrated in increasing alcohol concentrations prior to immersion in xylene for mounting. Slides were mounted with coverslips using DePeX mounting medium. Slides were scanned at 40× magnification using a Nanozoomer at the Histopathology Department, NUH and expression levels analysed using the NDPview2 software. Scoring was based on proportion (absent, <25%, 25–50% or >50%) of cells stained. A double-blinded approach was implemented in scoring to produce fair and reproducible results.

### 2.3. Cell Culture and Reagents

SF188 and KNS42 paediatric GBM cell lines were cultured in Dulbecco Modified Eagle’s Medium (DMEM)/F-12 nutrient mix (Gibco) supplemented with 10% Foetal Bovine Serum (FBS) and 2% L-glutamine. Adult GBM cell line, GIN28 was cultured in DMEM (Gibco; Milton Park, Oxford, Innovation Centre 99 Park Drive Milton Park, Oxford OX14 4RY, UK) containing 15% FBS, 1% L-glutamine and antibiotics (1% penicillin and streptomycin). HeLa cell line were used as positive control and grown in DMEM (Gibco) with 10% FBS and 2% L-glutamine. Cells were passaged at approximately 80% confluency by dissociating with trypsin-EDTA and seeded onto treated tissue culture plastic (polystyrene) and chamber slides. GBM cell lines were passaged twice a week to maintain exponential growth at varying seeding ratios of 1:3–1:5 and at cellular densities of 15,000–30,000 cells per well. HeLa cells were passaged daily due to their rapid doubling time and plated at 5000 cells per well. Optimal culture conditions of 37 °C, 21% Oxygen and 5% Carbon IV Oxide was maintained for all experiments.

### 2.4. Immunofluorescence (IF)

SF188, KNS42, GIN28 and HeLa cells were plated on 8-well chamber slides at optimised densities and incubation periods. Cells were washed in PBS, fixed in 0.4% paraformaldehyde for 20 min, permeabilised with PBS-T (0.1% tween, 0.25% triton-X in PBS) and blocked with 5% NGS in PBS-T for 1 h at room temperature. Primary anti-LDLR antibody was optimised on HeLa cells at dilutions ranging from 1:50–1:1000 and incubated overnight at 4 °C, subsequent experiments were performed at 1:250 dilution. Cells were incubated with goat anti-rabbit IgG secondary antibody labelled with Alexa Fluor 488 (Invitrogen, A11008) at 1:200 dilution for 1 h in the dark and washed in PBS-T. Slides were mounted with coverslips using DAPI-containing Vectastain mounting medium (Vector Labs Ltd., The Hauser Forum, 3 Charles Babbage Road, Cambridge, CB3 0GT, UK). Microscopy was carried out using a Nikon eclipse 90*i* fluorescent microscope and images were captured with Hamamatsu camera and Volocity 6.0 software at 40× magnification. ImageJ software was used to analyse images and correlate corrected total cell fluorescence (CTCF) with amount of LDLR protein expressed based on intensity of signal generated.

### 2.5. Statistical Analysis

All statistical analysis was performed using SPSS 24.0 software. Friedman ANOVA and Wilcoxon signed rank tests were used to analyse the significance of adult intra-tumour results. Mann–Whitney U or Kruskal–Wallis tests were used to determine the relationship between LDLR expression and clinicopathological factors. The overall survival time was compared using Kaplan–Meier and log-rank methods. Localisation of positive tumour cells within peri-vascular niches and LDLR expression in GBM cell lines were assessed with the One-way ANOVA test and Tukey’s HSD post hoc test for multiple comparisons, and *p* < 0.05 was considered statistically significant.

## 3. Results

### 3.1. LDLR Protein Is Expressed in a Majority of Adult GBMs with Intra-Tumour Variation

Patient diagnosis and clinico-pathological details of TMA samples are shown in Table 1.

LDLR protein expression was assessed in three distinct GBM intra-tumour regions representing core, rim, and invasive margin, from 36 patients. A universal expression was observed across all the regions, where 92% of samples from the tumour core had a frequency of positive cells above 25%. Similar results were also observed in the rim and invasive margins where more than 70% of cases expressed a frequency of LDLR positive cells greater than 25% Figure 1 and Table 2). Friedman ANOVA test showed a statistically significant difference among these intra-tumour regions (*p* = 0.008) and further pairwise analysis revealed that the observed significant differences were between the core and either rim or invasive margins (*p* = 0.003 and 0.014 respectively). No significant difference was detected in LDLR expression between the rim and invasive margin (*p* = 0.683, Wilcoxon’s signed rank test).

### 3.2. LDLR Is Expressed in Paediatric Malignant Gliomas Irrespective of Tumour Grade

The expression of LDLR protein in grade 3 (anaplastic astrocytoma) and grade 4 (GBM) paediatric samples (Figure 2) shows an expression pattern similar to that observed in adult GBM cases with 112 of the 133 samples (84% of cases) exhibiting expression levels above 25% (Table 3). Statistical analysis showed no significant difference in expression between grades 3 and 4 (*p* = 0.285, Mann–Whitney U test). This suggests a ubiquitous role for LDLR in high grade glioma metabolism, which could be exploited in developing novel future therapeutic strategies for both adult and paediatric anaplastic astrocytoma and glioblastoma cases.

### 3.3. Glioma LDLR Expression Is Localized within Peri-Vascular Niches

Positively stained tumour cells were repeatedly observed to localize preferentially in varying proportions near blood vessels compared to other areas of the tumour. We therefore examined the distribution of positive tumour cells in respect to vasculature in our adult and paediatric TMAs (Figure 3A–C). The average number of positively stained tumour cells around each identified vessel in paediatric samples was found to be 4.74 (95% CI: 4.48–5.00), while in adults it was 5.19 (95% CI: 4.84–5.84). However, intra-tumour variation was observed in the three distinct adult regions with the core having on average, the greatest number of positively stained tumour cells per vessel, while the invasive margin had the least (Table 4). Additionally, statistical comparisons among these regions by One-way ANOVA and Tukey HSD tests confirmed significantly higher numbers of tumour cells localized around blood vessels (*p* = 0.043, Figure 3D). This therapeutically relevant finding suggests that LDLR-positive tumour cells are more likely to be located within the penetration distance of nanoparticles upon extravasation from the bloodstream and will therefore potentially benefit from such future therapeutic targeting.

### 3.4. LDLR Expression Is Not Dependent on Clinicopathological Variables

We next assessed the relationship between LDLR expression and clinicopathological characteristics of known prognostic and predictive significance such as age, gender, tumour location, recurrence, and treatment. Statistical comparisons revealed that in both paediatric and adult cases, expression of LDLR generally did not correlate with these factors (*p* > 0.05, Mann–Whitney U and Kruskal–Wallis tests, with the exception of age in tumour core regions of in the adult cohort) (Table 5).

To determine the usefulness of LDLR expression as a predictor of survival as previous studies in breast and lung cancers have shown [22,23], we examined overall survival among our paediatric cohort (adult cases were excluded from analysis due to lack of information on survival for the majority of the cases). We observed that the median survival was lowest in cases with less than 25% expression and highest in those above 50% (5 and 11 months respectively); results were however not statistically significant (*p* = 0.126, Log-rank test) (Figure 4).

### 3.5. Adult and Paediatric GBM Cell Lines Express Variable Levels of LDLR

To ascertain that our cell lines expressed LDLR and were representative of results obtained from IHC of adult and paediatric TMAs, and therefore amenable to utilize as future in vitro models to test LDLR-targeted nanoparticle uptake/retention, expression levels were examined in three GBM cell lines. SF188, KNS42 (both paediatric) and GIN28 (adult). This revealed widespread staining confirming LDLR expression (Figure 5A). The corresponding amount of expression in the different cell lines determined by correlating corrected total cell fluorescence (CTCF) with integrated density based on the intensity of signal generated, showed that the SF188 and KNS42 paediatric GBM lines have relatively higher LDLR expression compared to GIN28 GBM cells (Figure 5B). Further analysis confirmed these differences were statistically significant (*p* = 0.001 and *p* = 0.008 for SF188 vs. GIN28 and KNS42 vs. GIN28 respectively, One-way ANOVA test) and on pair-wise comparisons, no significant difference was observed between the paediatric cell lines (*p* = 0.623, Tukey HSD test). This was in harmony with previous studies showing variable LDLR expression between different cell lines [17].

## 4. Discussion

As metabolic reprogramming is a well-established hallmark of cancer [28], research emphasis is shifting towards identifying altered metabolic pathways to serve as novel targets for anti-cancer drug delivery in order to maximize therapeutic benefits and efficiency [24]. Furthermore, increasing understanding of inter- and intra-patient genetic heterogeneity, renders common metabolic vulnerabilities particularly attractive for therapy, when considering the difficulties of applying personalized medicine to highly heterogeneous neoplasms. While the role of LDLR in maintaining tumour growth and proliferation has been well established in other cancers such as breast, lung, and gynaecological malignancies for decades, its role in malignant gliomas has not been extensively researched and evidence surrounding it is limited despite high grade gliomas being a leading cause of cancer related deaths worldwide [16,22,29]. LDLs, which are the principal carriers of cholesterol in circulation and required to support the growth of rapidly proliferating cancer cells, are the major ligands of LDLR, at least for canonical LDLR-based signalling. Abundance of LDLR thus signifies increased uptake and accumulation of cholesterol within the tumour and is an indicator of tumour aggressiveness and invasive potential as evident in high-grade gliomas [20,27,30,31].

We hypothesized that high-grade gliomas express invariably high levels of LDLR, despite the known genetic inter- and intra-tumour heterogeneity. Our results demonstrated widespread LDLR expression in the majority of the adult GBM cases with intra-tumour variation noted and a similar paediatric expression pattern which was independent of tumour grade, confirming LDLR as a valid therapeutic target across all ages for malignant glioma. Intriguingly, the localization of LDLR-expressing tumour cells within peri-vascular niches was observed in the vast majority of both adult and paediatric high-grade gliomas. Additionally, LDLR expression was discovered not to vary with clinicopathological factors, nor did it correlate with survival. However, a *p*-value of 0.126 (when comparing the <25% and >50% LDLR expression cohorts) certainly warrants re-examination in a larger cohort of high-grade glioma patients to refute a prognostic role more confidently for LDLR expression. Finally, we confirmed variable levels of detectable LDLR expression in adult (GIN28) and paediatric (SF188, KNS42) GBM in vitro cell lines, thus rendering these lines as future tools to test LDLR-targeted nanotherapy delivery.

The intra-tumour variation observed in adult GBM cases, where the core had the highest LDLR expression and the invasive margin the least, is consistent with current literature on intra-tumour heterogeneity [3,14]. The core is known to be the most cellular, proliferative and vascular region of the tumour compared to the periphery (which includes the rim and invasive areas) as demonstrated previously in our laboratory [32]. Previous studies have established mutually exclusive phenomena of proliferation and migration in tumours. The cells in the core are thus more proliferative, while those in the periphery/invasive margin are more migratory, invading into surrounding local tissue [33]. Although the invasive margin is the least proliferative, it is almost impossible to completely resect this infiltrative region during surgery. As a direct consequence, residual invasive cells left behind in this region are thought to be responsible for the high degree of tumour recurrence and high mortality [34]. Based on our findings, it is thus logical to assume that LDLR expression at the invasive margin represents a more therapeutically relevant population of cells amenable to LDL-mediated nanoparticle/drug targeting, which could potentially and significantly reduce recurrence rates and improve patient outcome. Furthermore, it also suggests that in cases where extensive tumour debulking is not possible, cells in the core expressing a higher percentage of LDLR could benefit from such targeting as an alternative form of intervention.

The non-significant difference in levels of LDLR expression between anaplastic astrocytoma and glioblastoma (*p* = 0.285) in our paediatric cohort points to a possible ubiquitous role of LDLR in paediatric high grade glioma metabolism. Based on this finding, we can predict that both anaplastic astrocytoma (grade 3) and glioblastoma (grade 4) may be amenable to anti-LDLR targeting to the same extent. However, re-examination in a larger paediatric cohort is required to validate a claim of LDLR as a broadly ubiquitous target. Considering that children are not able to tolerate the same levels of intense treatments as adults and that current standard-of-care treatment modalities, such as radiation therapy, are exempt in children under 3 years of age, paediatric patients might potentially benefit more from preferential LDLR targeting. Although our data did not show paediatric LDLR expression to be dependent on tumour grade, differences may be more subtle and require a larger sample size to interrogate.

LDLR expressing tumour cells were consistently observed to localize within perivascular areas, consistent with seminal studies by Calabrese et al., in 2007, across a host of primary brain tumours, notably glioblastoma, ependymoma, medulloblastoma and oligodendroglioma, where cancer stem cells (CSCs) were shown to associate in close proximity to tumour capillaries creating vascular niches. This tumour microenvironment is essential for the maintenance of tumour growth and proliferation and is one of the key rationales for several anti-angiogenic therapies [35]. The overall significant difference in vascularity detected across adult GBM intra-tumour regions was essentially due to the high vascularity of tumour cells in the core compared to the invasive margin (*p* = 0.040) as confirmed by Tukey’s HSD test. This phenomenon can be explained by the higher cellular density and proliferation of the core compared to other regions which outstrips it of its blood supply, inducing a state of nutrient deficiency and hypoxia—the main driver of angiogenesis and neo-vascularization within the tumour. Similar comparison between paediatric anaplastic astrocytoma and glioblastoma was limited by disparity in sample sizes (13 and 55 respectively) which would have given a biased result. A larger sample size, particularly of grade 3 tumours, would have improved the power and validity of such analysis.

The association of positive tumour cells with the vasculature is of immense therapeutic significance. Hitherto, one of the greatest challenges of anti-cancer drug delivery to tumour sites is the difficulty in penetrating and accumulating to sufficient levels as to effect cell kill within the target organ, without incurring significant toxicity to surrounding normal tissues. In attempts to overcome this, novel drug delivery strategies such as macromolecular drug carriers have been developed [36,37]. A study by Dreher i.e., in 2006, showed that lower weight macromolecular drug carriers and liposomes have a higher vascular permeability and longer plasma half-life due to enhanced permeability and retention effects [38]. This in effect indicates that LDLR-expressing tumour cells within close proximity to the blood vessels as shown in our results, are potentially more susceptible to efficacious therapeutic targeting using LDL-mimicking stealth nanoparticles. In light of our data, and for future research into this aspect, we hypothesise that effective nanoparticle-mediated tumour-cell kill is a function of peri-vasculature localization. Furthermore, the use of statins has been reported to reduce the efflux activity of drug transporters Pgp/ABCB1 and BCRP/ABCG2, increase the number of LDLR exposed on the BBB surface, thereby increasing doxorubicin-loaded nanoparticle entry across the BBB [39]; polyester-based nanoparticles loaded with paclitaxel improved BBB permeability in vitro [40]; and an infiltrative GBM preclinical model has been recently shown to be sensitive to enhanced proton treatment with LDLR-ligand peptide-conjugated gold nanoparticles [41].

## 5. Conclusions

Taken together, our results have clearly shown the LDLR pathway to represent a metabolic vulnerability in both paediatric and adult high-grade gliomas, which potentially circumvents intra- and inter-tumour heterogeneity and is therefore potentially amenable to future lipoprotein based anti-LDLR nanotherapy, ubiquitously in patient cohorts. To maximize the benefits from this novel therapeutic strategy, there may be a need for selection of cases likely to benefit the most based on demonstrated levels of LDLR expression, tumour vascularity and/or further lipidomic studies.

## Figures and Tables

**Figure 1 pharmaceutics-15-00599-f001:**
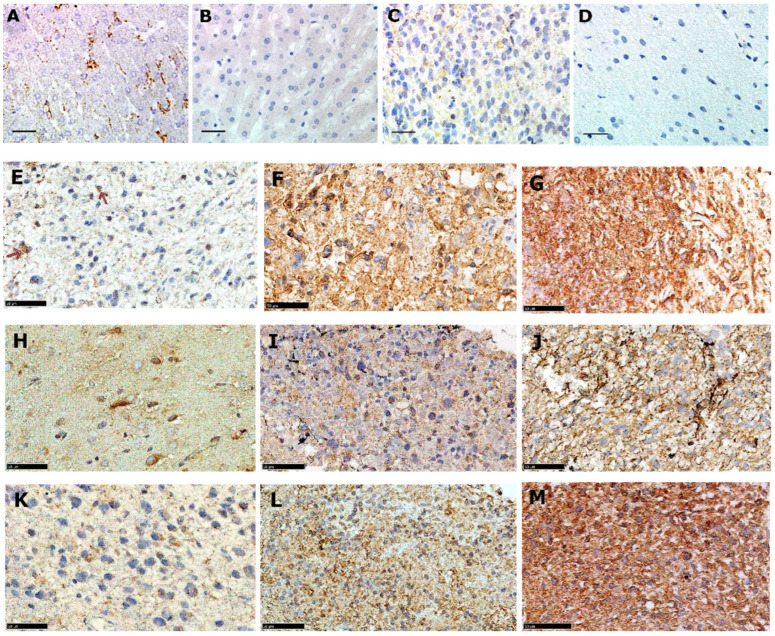
Anti-LDLR immunohistochemical analyses of adult GBM primary tissue. (**A**) Positive liver control. (**B**) Negative liver control. (**C**) Positive normal brain control. (**D**) Negative normal brain control. (**E**) Core with <25% expression. (**F**) Core with 25–50% expression. (**G**) Core with >50% expression. (**H**) Rim with <25% expression. (**I**) Rim with 25–50% expression. (**J**) Rim with >50% expression. (**K**) Invasive margin showing <25% expression. (**L**) Invasive margin with 25–50% expression. (**M**) Invasive margin with >50% expression based on proportion of positive cells. All images were captured at 40× magnification; scale bars correspond to 100 µm in (**A**–**D**) and 50 µm in (**E**–**M**).

**Figure 2 pharmaceutics-15-00599-f002:**
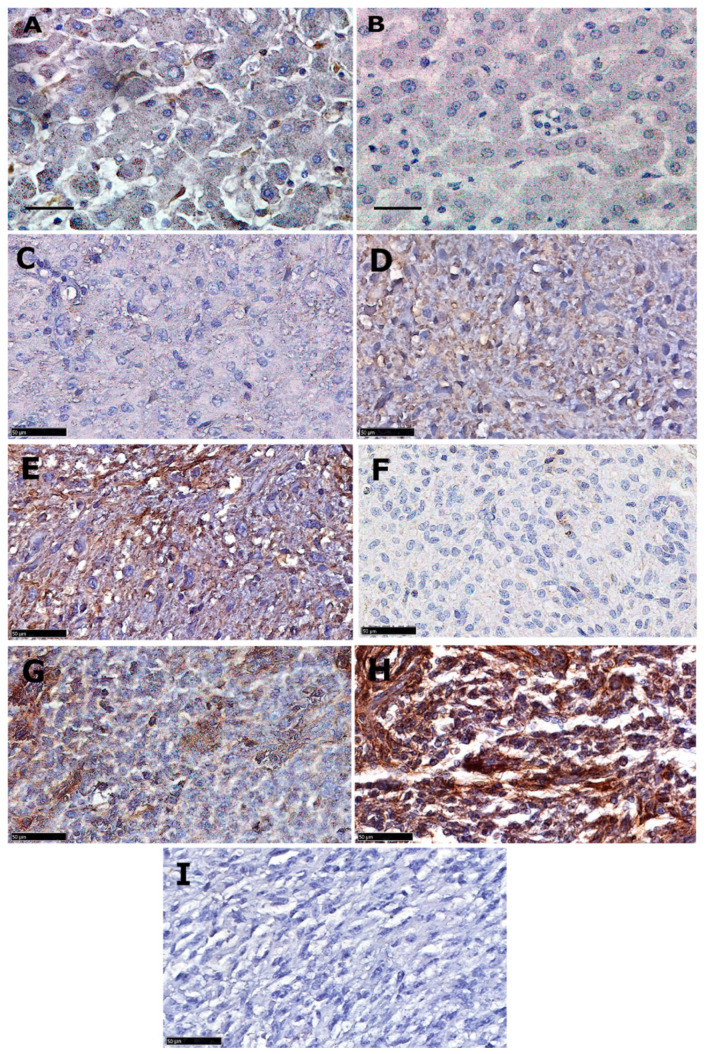
Anti-LDLR immunohistochemical analyses of paediatric high-grade glioma. (**A**) Positive liver control. (**B**) Negative liver control. (**C**) Anaplastic astrocytoma (AA, grade III) cells showing less than 25% expression. (**D**) AA showing 25–50% expression. (**E**) AA with >50% expression. (**F**) GBM (grade IV) cells with less than 25% expression. (**G**) GBM cells with 25–50% expression. (**H**) GBM cells with >50% expression. (**I**) GBM cells with negative protein expression. All images were captured at 40× magnification; scale bars correspond to 100µm in (**A**,**B**) and 50 µm in (**C**–**I**).

**Figure 3 pharmaceutics-15-00599-f003:**
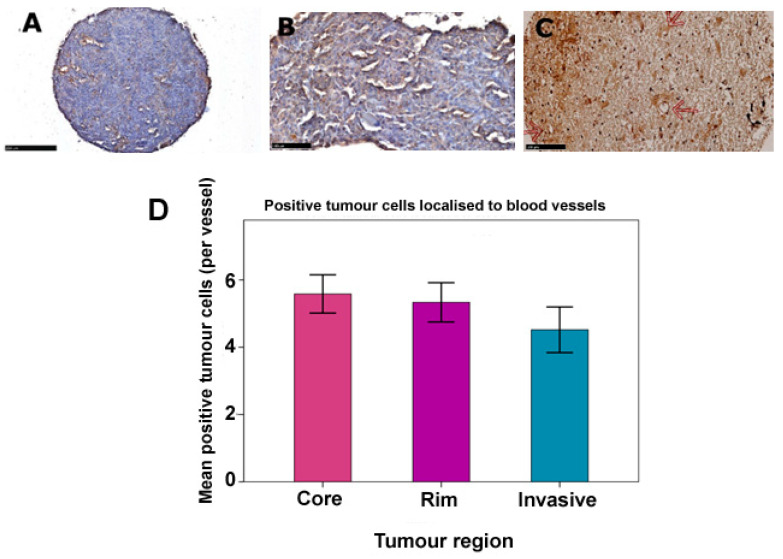
Localization of LDLR positive tumour cells in peri-vasculature regions. (**A**) Paediatric GBM whole tumour core showing preferential localisation of LDLR expression (brown staining) around blood vessels compared to other areas. (**B**) Highly predominant localisation of LDLR-expressing paediatric GBM cells around the vasculature. (**C**) Invasive region of adult GBM showing tumour cells with positive LDLR staining in perivascular areas (indicated by red arrows). Image magnification; A, 10×, B and C, 20×. scale bars correspond to 250 µm for A and 100 µm for B and C. (**D**) Variation in average number of positive tumour cells per vessel in adult intra-tumour regions (*p* = 0.043), where pair-wise analysis revealed a significant difference in core and invasive regions only (*p* = 0.040, Tukey HSD test).

**Figure 4 pharmaceutics-15-00599-f004:**
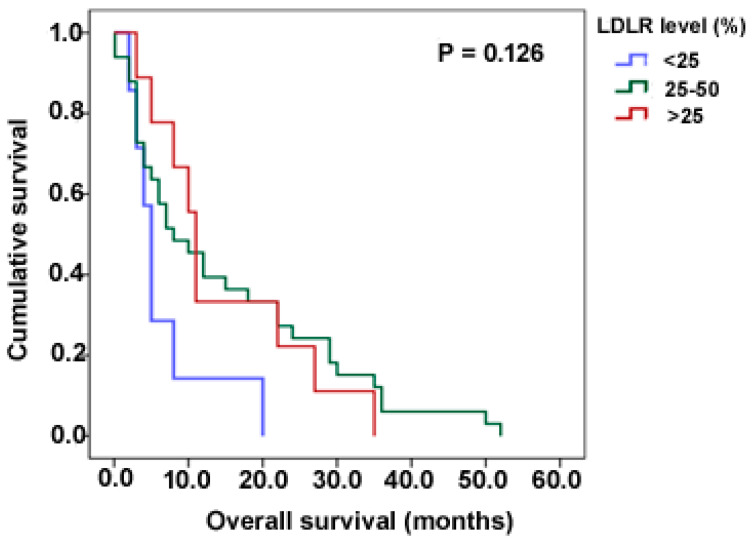
Association of LDLR expression with clinico-pathological characteristics. Kaplan–Meier plot showing median overall survival time of 5, 8, and 11 months with LDLR expression levels of <25%, 25–50% and >50% respectively in paediatric high-grade gliomas. Statistical analyses were not significant, indicating no correlation between overall survival and LDLR expression (*p* = 0.126).

**Figure 5 pharmaceutics-15-00599-f005:**
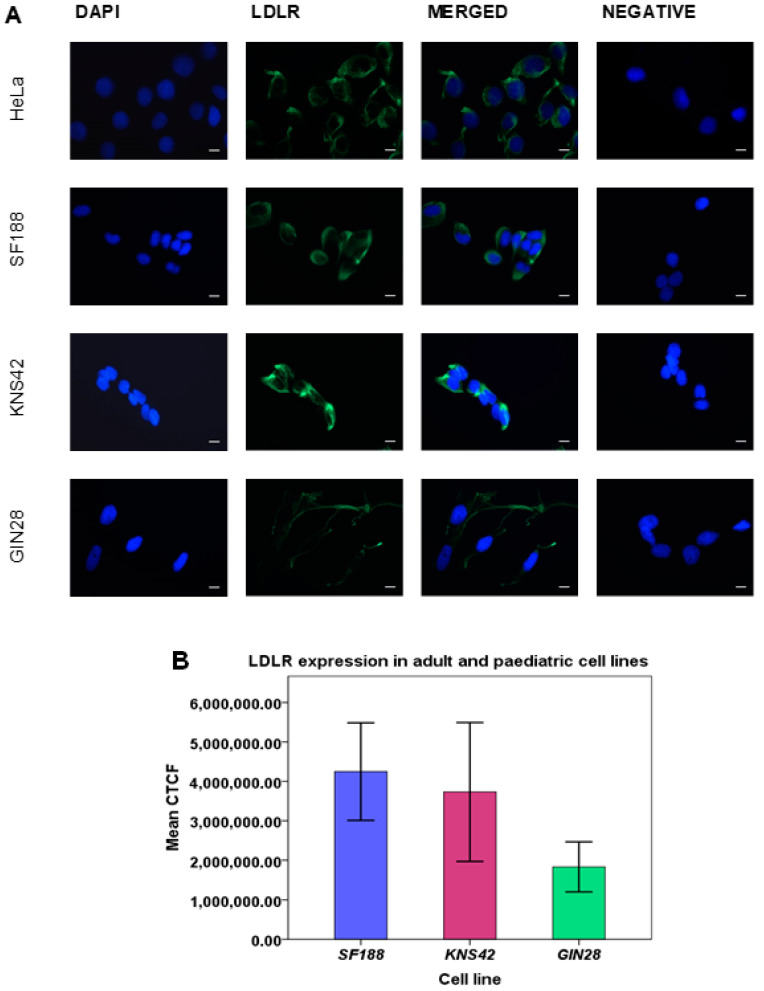
In vitro analysis of LDLR expression in adult and paediatric GBM cell lines. (**A**) Immunofluorescence analysis showing LDLR expressions in control HeLa cell line, paediatric cell lines SF188 and KNS42, and adult cell line GIN28. Blue—nuclear stain DAPI; Green—LDLR protein. Magnification 40×; scale bar: 100 µm. (**B**) Mean LDLR expression in all cell lines showing highest expression in SF188 and lowest in GIN28 (*p* = 0.001), where error bars correspond to 95% CI.

**Table 1 pharmaceutics-15-00599-t001:** Clinicopathological characteristics of adult and paediatric high grade glioma cases.

Variable	Adult ^a^ Frequency (%)	Paediatric ^b^ Frequency (%)
Age (years) ^c^		
paediatric ^d^		29 (39.2)
<6		30 (40.5)
6–12		15 (20.3)
>12		
Adult	7 (19.4)	
<50	16 (44.4)	
50–65	13 (36.1)	
>65		
Gender ^d^		
Male	18 (50)	53 (64.6)
Female	18 (50)	29 (35.4)
Tumour grade ^d^		
III (AA)	-	13 (19.1)
IV (GBM)	36 (100)	55 (80.9)
Tumour location ^d^		
Supratentorial		
Infratentorial	36 (100)	57 (73.1)
Treatment (post-surgery) ^d^	-	21 (26.9)
Radiotherapy		
Chemotherapy	2 (12.5)	9 (18.8)
Chemotherapy + Radiotherapy	1 (6.3)	15 (31.3)
None	12 (75)	23 (47.9)
Survival time (months) ^d,e^	1 (6.3)	1 (2.1)
<12		
12–36	5 (100)	29 (63)
>36	-	15 (32.6)
Tumour type ^d^	-	2 (4.3)
Primary		
Recurrent		
progressive	N/A	66 (84.6)
	N/A	10 (12.8)
	N/A	2 (2.6)

^a^ Total number of cases—36. ^b^ Total number of cases—133. ^c^ Paediatric mean age 7.7 years (95% CI: 6.6–8.8), adult median age 59 years (interquartile range: 53.25, 67.75). ^d^ Missing details for some cases. ^e^ Median survival time; adults—4 months (interquartile range: 3, 6), paediatric—8 months (interquartile range: 3, 22). N/A—Not available.

**Table 2 pharmaceutics-15-00599-t002:** LDLR expression in distinct intra-tumour regions of adult GBM.

% Expression	Core	Frequency (%) Rim	Invasive
Negative	0	0	0
<25	3 (8.3)	8 (22.2)	8 (25.8)
25–50	14 (38.9)	19 (52.8)	15 (48.4)
>50	19 (52.8)	9 (25)	8 (25.8)
Total	36 (100)	36 (100)	31 (100) *

* Invasive margin samples not available for five patients.

**Table 3 pharmaceutics-15-00599-t003:** LDLR expression in paediatric anaplastic astrocytoma and GBM.

% LDLR Expression	Frequency (%)
Negative	5 (3.8)
<25	16 (12)
25–50	77 (57.9)
>50	35 (26.3)
Total	133 (100%)

**Table 4 pharmaceutics-15-00599-t004:** Categorical proportions of positive tumour cells per vessel in adult and paediatric cases and corresponding *p*-values.

LDLR Expression ^a^	Core	Frequency (%) Adult Rim	Invasive	Paediatric ^b^
Low	1 (2.8)	4 (11.1)	8 (27.6)	22 (18.0)
Moderate	24 (66.7)	23 (63.9)	17 (58.6)	86 (70.5)
High	11 (30.6)	9 (25.0)	4 (13.8)	14 (11.5)
Total	36 (100)	36 (100)	29 (100)	122 (100)

^a^ Number of LDLR positive cells around each identified blood vessel. Low < 4, moderate 4–6, high > 6. ^b^ Comparison between anaplastic astrocytoma and GBM was limited by lack of clinical diagnosis for the majority of cases.

**Table 5 pharmaceutics-15-00599-t005:** Statistical analyses of LDLR expression and clinico-pathological characteristics.

Variable	Core	Adult Rim	Invasive	Paediatric
Age	0.047	0.653	0.578	0.445
Gender	0.106	0.223	0.623	0.270
Tumour site ^a^	0.382	0.173	0.462	0.846
Treatment	0.186	0.173	0.362	0.431
Tumour type ^b^	-	-	-	0.958

^a^ Supra-vs. infra-tentorial location. In adults, where all cases were supratentorial, differences were examined amongst frontal, parietal, temporal and occipital tumour sites. ^b^ Primary or recurrent. No clinical information available for adult cases.

## Data Availability

Data are contained within the article.

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
