# Peer review of "Low-Density Lipoprotein Pathway Is a Ubiquitous Metabolic Vulnerability in High Grade Glioma Amenable for Nanotherapeutic Delivery"

_pharmaceutics, 2023, doi:10.3390/pharmaceutics15020599_

Round 1

Reviewer 1 Report (New Reviewer)

The manuscript is well written and the information is well structured. The work is original and the approach is very interesting.
The results are consistent with the hypothesis, but Figure 5 does not show significant results and shows weaknesses in the work. They should include results with significant differences to support that result and draw some clear conclusion. Failing that, back it up with another test.
They should include more current bibliography, it is too obsolete.

Author Response

Reviewer 2 Report (New Reviewer)

The submitted manuscript, entitled -Low density lipoprotein pathway is a ubiquitous metabolic vulnerability in high grade glioma amenable for nanotherapeutic delivery- is an excellent example how import interdisciplinary research activities are to investigate more efficient anti-cancer therapies.

The authors clearly address the need for new therapeutics against glioblastoma in general but although potential treatment targets for adults and children. Especially the paediatric focus distinguishes this study from many others and underline the relevance of this study.

The whole manuscript is comprehensively structured  and professionally explains the introduced topic.

Minor comments:

1.) In general, there are different, potential “drug conjugates” applicable and maybe the term “nanotherapeutic” does not cover them all. Due to the fact that the authors do not focus on the development of certain therapeutics, they could address possible strategies by referring more publications within the last five or ten last years.

2.)Additionally, we analysed expression levels in three representative cell line models to confirm their future utility to test LDLR-targeted nanoparticle uptake, retention, and cytotoxicity­- (line 15 but also in the discussion and conclusion part), in this respect the authors should emphasize that meanwhile different nanoparticulate formats have been tested (e.g. PMID: 22788770; doi: 10.1111/j.1476-5381.2012.02103; The association of statins plus LDL receptor-targeted liposome-encapsulated doxorubicin increases in vitro drug delivery across blood–brain barrier cells; ML Pinzón-Daza, et.al.; Br J Pharmacol. 2012 Dec; 167(7): 1431–1447) and what is the novelty in their study.

Author Response

This manuscript is a resubmission of an earlier submission. The following is a list of the peer review reports and author responses from that submission.

Round 1

Reviewer 1 Report

The authors have determined LDL receptor (LDLR) expression in 12 high grade gliomas using immunohistochemistry on tissue microarrays from intra- and inter tumour regions of 36 adult and 133 paediatric patients. The purpose was to confirm expression level and location of LDLR as a therapeutic target. For the manuscript, reviewer has some concerns for publication and provide the improvement point of this manuscript as follows:

Major:

L319, Figure 5: This data could be data only for understanding the situation of expression of LDLR. It is difficult to regard them as new discoveries or scientific contributions. The author should show specific application examples, such as LDLR-targeted nanotherapy, using these celllines to make sure that an applications using them are possible.

L372-374: The authors insisted “The non-significant difference in levels of LDLR expression between anaplastic astrocytoma and glioblastoma (p=0.285) in our paediatric cohort points to a possible ubiquitous role of LDLR in paediatric high grade glioma metabolism.” in Discussion part. The p value 0.285 was not so big. If a large number of samples were analyzed, there might be significant difference between them. In this case, it could be risky to explain the logic of the sentence using this p value and no significance.

Figure 4: The result showed that middle LDL levels is long overall survival. Please explain the interpretation more clearly. Also, p = 0.126 is difficult to think as no correlation. The p = 0.126 means no correlation but the result might be due to low number of samples. More analysis using sample may be needed in this experiment.

Minor:

Line 408: There is no need to use the abbreviation for enhanced permeability and retention effects as EPR. Please delete “(EPR)”.

Figure 4: Please unify probability to P or p in figure and text.

Reviewer 2 Report

This research article is aimed at characterizing the expression of LDLR in high grade glioma. The manuscript is well written and the experimental design is well conducted. However, the novelty is severly undermined by other published papers demonstrating that LDLR is generally intensely expressed in glioma cells. Additionally, the authors excessively stress the association to LDLR-targeted nanotherapies. However, they do not show any proof of evidence about their effectiveness. This is only a characterization of LDLR expression in glioma cells/tissues, which partly overlap with what already known in other published studies (Wood et al., 2022; Maletinska et al., 2000; Pizzocri et al., 2021Tworowska et al, 2022)